# Socio-Demographic and Clinical Characteristics of Patients with Substance Intoxication Receiving a Psychiatric Assessment in the Emergency Department of the Maggiore Della Carita Hospital, Novara, Italy

**DOI:** 10.3390/ijerph22010023

**Published:** 2024-12-28

**Authors:** Eleonora Gambaro, Carla Maria Gramaglia, Davide Cenci, Daniela Ferrante, Francesco Gavelli, Mattia Bellan, Patrizia Zeppegno

**Affiliations:** 1Psychiatry Unit, Maggiore della Carità Hospital, 28100 Novara, Italy; carla.gramaglia@med.uniupo.it (C.M.G.); davide.cenci@maggioreosp.novara.it (D.C.); patrizia.zeppegno@med.uniupo.it (P.Z.); 2Department of Translational Medicine, Università del Piemonte Orientale, 13100 Vercelli, Italy; francesco.gavelli@uniupo.it (F.G.); mattia.bellan@med.uniupo.it (M.B.); 3Department of Translational Medicine, Università del Piemonte Orientale, SSD Epidemiologia dei Tumori, AOU Maggiore della Carità e CPO Piemonte, 28100 Novara, Italy; daniela.ferrante@uniupo.it; 4Emergency Medicine Department, Maggiore della Carità University Hospital, 28100 Novara, Italy

**Keywords:** alcohol intoxication, substance use, emergency department, psychiatric assessment

## Abstract

Patients intoxicated with alcohol or other substances are often assessed and assisted in the context of the Emergency Department (ED) by emergency physicians, who usually require a psychiatric assessment. This study aims to analyse the characteristics of a sample of patients receiving a psychiatric consultation in the ED setting of the Maggiore della Carità University Hospital in Novara, Italy, in the period from 1 January 2021 to 31 December 2023, to find out whether and how patients screening positive for alcohol/drugs differ from those screening negative. Socio-demographic and clinical history information and clinical data related to the ED psychiatric consultations were extracted from the PsNet database, a collection of data extracted from the application that serves as the electronic medical record for patients accessing the ED. Statistical analyses were performed using SAS 9.4 and STATA version 17 software. Chi-square/Fisher tests, *t*-tests, and both univariate and multivariate logistic models were employed. Most of the findings regarding socio-demographic characteristics, symptoms observed during the 1362 psychiatric consultations, and treatments received by a total of 922 patients in the ED were consistent with the literature on this topic. However, some results only partially aligned with previous studies, particularly concerning the higher frequency of anxiety and psychotic symptoms, as well as cognitive impairments, observed in consultations for patients who tested negative for alcohol or substances. Interpreting these findings is complex and raises important questions, which may be addressed more effectively by expanding the sample size (extending the research to other EDs) and analysing its characteristics in greater detail. In this regard, improving diagnostic methods for detecting substance use through laboratory tests would also be beneficial.

## 1. Introduction

Substance abuse represents a problem with significant medical and social impact. Intoxication or withdrawal from drugs is often the cause of acute symptoms that lead patients to seek care in the Emergency Department (ED) [1].

Understanding the characteristics, effects, and risks associated with the abuse of various substances is essential to counteract potentially lethal outcomes, provide appropriate care during the acute phase of intoxication or withdrawal, and refer patients to a proper outpatient or inpatient detox programme [2,3,4].

Substance abusers who access the ED often present with a combination of physical, psychological, and social issues, sometimes severe agitation [4,5,6,7]. This calls for a multidisciplinary intervention, potentially including psychiatric evaluation [8].

The increasing prevalence of substance use disorders (SUDs) dramatically affects psychiatric health systems globally, particularly in ED.

Globally, substance use continues to rise at alarming rates. According to the World Health Organisation (WHO), approximately 35 million people worldwide suffer from drug use disorders, with many experiencing co-occurring mental health conditions [9].

Substance intoxication often leads individuals to seek emergency medical assistance, particularly in EDs where the need for immediate psychiatric evaluation is essential for effective intervention. The socio-demographic characteristics of these patients can vary significantly, with noticeable trends among different regions. Factors such as age, sex, education level, and socio-economic status often inform substance use patterns and associated health outcomes [10].

International data show varying trends in substance use patterns influenced by cultural and legal contexts. In Europe, countries such as the Netherlands and Portugal have adopted progressive drug policies, which have led to different patterns of substance use and treatment outcomes [9]. Nonetheless, EDs in these regions still encounter patients with serious intoxication-related psychiatric needs. The socio-demographic characteristics of these patients often include a diverse age range, with increasing reports of older adults facing complications from long-term substance use. Global health trends indicate a rising prevalence of dual diagnosis, where individuals present with both substance use disorders and mental health conditions, necessitating integrated approaches in emergency care settings [11,12]

In Sub-Saharan Africa, the double burden of infectious diseases and escalating substance use adds complexity to healthcare systems. Recent studies [13,14] indicate rising rates of alcohol use and an increasing presence of illicit drugs, such as cocaine and methamphetamines. Reports suggest that the socio-demographic profile of affected individuals typically includes young adult males, with substantial variation in prevalence among urban versus rural populations. Cultural norms and socio-economic factors play a crucial role in substance use trends; individuals from lower socio-economic backgrounds often experience higher rates of addiction and co-occurring mental health issues, exacerbating the challenges faced in accessing timely psychiatric care.

In the United States, the impact of substance intoxication on psychiatric assessments in EDs reveals significant strains on healthcare resources. For instance, the Substance Abuse and Mental Health Services Administration (SAMHSA) reported that a substantial proportion of emergency department visits involve patients with substance use disorders. Demographically, these patients often present as predominantly young males, although there is an increasing prevalence of women, especially among prescription drug users. Additionally, those with lower educational attainment and limited health literacy frequently face greater barriers to accessing appropriate care and follow-up services [15].

Substance use in Italy has been characterised by various patterns, with alcohol, cannabis, and cocaine being the most abused substances. According to the Italian National Drug Report 2021, approximately 10.5% of the Italian population aged 15 to 64 reported using illegal substances at least once in their lives [16].

Furthermore, the National Institute on Drug Addiction (Istituto Superiore di Sanità, ISS), highlighted an increase in emergency department visits related to substance use, signalling the urgency for psychiatric assessments in these acute care settings [17].

In Italy, the rising incidence of substance use disorders (SUDs) calls for an enhanced understanding of the socio-demographic and clinical profiles of individuals presenting with substance intoxication. This understanding is crucial for developing effective intervention strategies and resource allocation within healthcare systems [18].

Research indicates that the socio-demographic characteristics of patients with substance intoxication in Italy reveal significant patterns. Most patients presenting with substance intoxication in EDs are young adults, typically aged 18 to 34 [18]. This trend reflects not only developmental risk factors but also the accessibility of substances to this age group. Men account for a substantial majority of substance intoxication cases, often suggested to be influenced by societal factors and risk-taking behaviours. Women are increasingly represented in these cases, particularly concerning the misuse of prescription drugs and alcohol. Patients from lower socio-economic backgrounds are disproportionately affected by substance use disorders. Economic stressors and lack of access to mental health resources are contributing factors leading to higher rates of substance-related emergencies [18,19].

Substance use trends vary significantly between urban and rural areas in Italy, with higher rates of reported intoxication cases in metropolitan regions, such as Rome and Milan. This discrepancy underscores the role of availability and urban culture in substance use patterns [17].

The clinical presentation of patients with substance intoxication in Italian EDs can vary widely based on the type of substance involved. Common clinical characteristics include the following: (1) Polydrug Use: It is increasingly common for patients to present with polydrug intoxication, complicating assessment and treatment. Reports indicate that many individuals combine alcohol with other substances, particularly cannabis and stimulants like cocaine; (2) Mental Health Comorbidities: A significant proportion of patients receiving psychiatric assessment display co-occurring mental health disorders, including anxiety and depression. This dual diagnosis can complicate treatment and often necessitates a more integrated approach to care; (3) Withdrawal Symptoms: Patients presenting with severe intoxication may also exhibit withdrawal symptoms, particularly in cases of habitual substance use. This aspect necessitates prompt intervention to address immediate safety concerns and longer-term rehabilitation needs; (4) Intent and Context of Use: understanding the intent behind substance use—whether recreational, self-medication for underlying mental health issues, or social—can inform psychiatric assessments and guide appropriate treatment options [20].

The socio-demographic and clinical characteristics of patients with substance intoxication receiving psychiatric assessments in Italian emergency departments illuminate a complex interplay of factors contributing to the current crisis. Young adults, predominantly males, from lower socio-economic backgrounds are particularly vulnerable, with many presenting with co-occurring mental health disorders. The trend in poly-drug use complicates clinical management and requires comprehensive treatment strategies [21].

Addressing substance use and its associated mental health implications in Italy necessitates a coordinated response involving public health initiatives, policy reforms, and an integration of services between emergency care and mental health support. Continued research and data collection are essential to inform these strategies and ensure effective intervention for those affected by substance intoxication in Italy [22]. Any hospital service can request a psychiatric consultation from the Psychiatric Diagnosis and Care Service (PDCS) for diagnostic assessments in patients exhibiting psychomotor agitation or suspected psychiatric pathology [23]. More specifically, as defined by the most recent edition of the Diagnostic and Statistical Manual of Mental Disorders (DSM-5), agitation is described as “the inability to sit still, pacing, handwringing; or pulling or rubbing of the skin, clothing, or other objects” and as “disruptive motor or vocal activity” [24]. Numerous scientific evidence underlines that acute agitation has multiple causes, which include not only medical conditions (like hypoglycemia, pain, and delirium) but also substance use (from alcohol or drugs) and psychiatric issues [25,26,27].

The ED often serves as the primary resource for the acute management of patients experiencing alcohol or substance abuse intoxication. This is the setting where initial care is provided, diagnostic processes are initiated, first psychiatric contact may take place (especially when patients show behavioural disorders and/or psychomotor agitation), and patients can eventually be referred to the most appropriate care pathways, either inpatient or outpatient. Therefore, a better understanding of this topic is warranted, as substance abusers with complex needs may represent a significant burden for the ED [8,28,29].

The purpose of this work is to examine psychiatric consultations in the ED setting during each period, to determine whether and how patients screening positive for alcohol/drugs differ from those screening negative. To achieve this, we analysed patients receiving a psychiatric consultation in the ED of the Maggiore della Carità University Hospital in Novara, Italy from 1 January 2021 to 31 December 2023.

## 2. Materials and Methods

This study is observational, cross-sectional, and non-profit. The data will be collected anonymously. It does not imply any change in normal clinical practice. For this study, all psychiatric consultations conducted at the ED of the Maggiore della Carità University Hospital in Novara, Italy from January 2021 to December 2023 were examined.

The only exclusion criterion applied was an age below 16 years, as this age group is under the care of Child Neuropsychiatry and typically accesses a separate pediatric emergency department.

During each psychiatric consultation conducted in the ED, a clinical evaluation was performed by a psychiatrist through a clinical interview. All necessary information for the study was extracted from the “PsNet” database, a collection of data extracted from the application that serves as the electronic medical record for patients accessing the ED. This database includes the following data for each patient who accesses the ED.: entry codes, demographic information, recent and remote pathological history, the evolution of patients’ clinical condition during ED stays, laboratory analyses, consultations, instrumental and imaging investigations performed, and type of discharge from the ED.

After each consultation, a datasheet was completed within an Excel database to collect information regarding the sociodemographic and clinical characteristics related to both the patients’ history and the consultation received in the ED:Reason for access and priority code;Demographic data: age, gender, nationality, residence, marital status, education, occupation, and information related to the mode of access;History of psychiatric hospitalisations and prior access to Drug Addiction Service;Psychiatric diagnosis according to DSM-5 and psychopharmacological therapy;Organic comorbidities and medical therapy;Dependencies on alcohol, substances, or behaviours;Blood alcohol level and detection of substances in urine (barbiturates, benzodiazepines, amphetamines, cannabinoids, opioids, cocaine);Suicidal ideation and self-harming behaviour;Acute therapy administered, main symptomatology, and outcome of the Intervention.

Data were recorded appropriately and anonymously in the specific database. There are no diagnostic evaluations or additional visits for this study.

Ethical considerations:

This study was conducted following the Declaration of Helsinki and approved by the Comitato Etico Territoriale Interaziendale of the AOU Maggiore della Carità di Novara (Prot. n°912/CE of the 3 July 2023) and Università del Piemonte Orientale (UNIUPO) as part of the research duties of the Psychiatry Institute. Informed consent was obtained from patients.

Statistical Analysis:

Separate descriptive statistical analyses were conducted for subjects with and without acute alcohol and/or substance intoxication (determined through blood or urine tests). Absolute and relative frequencies are reported for categorical variables, while mean, standard deviation, or median and interquartile range are reported for numerical variables. Chi-square/Fisher tests and *t*-tests or non-parametric alternatives were used to calculate differences in proportions between the groups.

Variables defined as “constant” over time, such as gender, age, residence, and education, were calculated using subjects as the denominator. For other variables, the number of consultations was used as the denominator. For gender, age, residence, and education, information from the first contact was considered; if data were missing, values from subsequent consultations were evaluated.

Univariable and multivariable logistic models were performed to evaluate the association between the characteristics of patients with alcohol or substance intoxication and those without, considering socio-demographic variables (gender, age, residence, marital status, education) and aspects of recent and past psychiatric history (previous psychiatric conditions, hospitalisations, patient under psychiatric care, previous contacts with the drug addiction service (SER.D.), patient under SER.D. care, presence of organic comorbidities, and psychopharmacological therapy), with a confidence interval set at 95%.

Analyses were performed using a stepwise approach, and statistical analyses were conducted using SAS 9.4 [30] and STATA version 17 [31]. The significance threshold was set at 0.05

## 3. Results

We analysed data from 1362 psychiatric consultations conducted at the ED of the Maggiore della Carità University Hospital in Novara, Italy, from January 2021 to December 2023, involving a total of 922 patients. The age recorded in 1348 consultations ranged from 16 to 98 years, with an average age of 45.58 years and a standard deviation (SD) of 18.463. In 1043 consultations, substance use was assessed, showing a positive result in 43.24% (N = 451) and a negative result in 56.76% (N = 592). Blood alcohol levels were requested for 945 consultations, with a blood alcohol concentration greater than 0.50 ng/mL found in 11.11% (N = 105) of patients and negative results in 88.89% (N = 840). A cross-check for alcohol and substance use was conducted in 887 consultations: in 51.63% (N = 458) of cases, patients were negative for both, while 48.37% (N = 429) tested positive for one or both (338 positive for substances but not for alcohol, 48 positives for alcohol but not for substances, and 43 positives for both alcohol and substances).

Statistically significant results obtained from the comparison between the two groups (N = 458 consultations for patients who tested negative for alcohol/substances; N = 429 consultations for patients who tested positive for alcohol/substances) are shown in the following Tables (Tables 1–5).

## 4. Discussion

We compared consultations for patients who tested negative for alcohol/substances in the ED setting to those who tested positive in all the variables collected and described above in the methods section, with a particular focus on psychiatric history and potential substance or behavioural dependencies.

### 4.1. Sociodemographic and Clinical Characteristics

Regarding socio-demographic and clinical information, a statistically significant difference was found between the two genders: among women, the patients who tested negative for alcohol/substances were more numerous than positives, while positive men were more numerous than negatives (Table 1).

The distribution was balanced between genders. Concerning this latter finding, previous studies have shown a predominant alcohol consumption among males; however, this gap has been narrowing over time, with an increasing number of women exhibiting Alcohol Use Disorder (AUD) and related pathological conditions [32,33].

The consultations for patients who tested negative or positive for alcohol/substances revealed a consistent distribution regarding their living arrangements: a statistically significant difference emerged in the context of consultations involving patients who were homeless, with a higher prevalence among those who tested positive for alcohol/substances. This finding may be linked to a higher rate of substance abuse among individuals in conditions of social marginalisation, who are acknowledged to be more likely to experience traumatic events and psychiatric disorders [34].

A higher percentage of consultations was found for patients who tested negative and were being followed for psychiatric reasons. Moreover, as expected, more consultations were recorded for patients who tested positive and were being followed at the SER.D (Table 1). These data contrast with the literature that recognises a frequent overlap between alcohol/substance abuse and psychiatric disorders, with percentages higher than the general population [35,36]; furthermore, numerous substances can lead to manifestations of a psychotic nature [37].

### 4.2. Ongoing Therapy Before the Consultation

Our study also investigated the types of therapies patients were taking at the time of consultation, revealing a significant difference for long-acting injectable (LAI) typical neuroleptics, which were more frequently present in the treatment regimens of patients who tested negative for alcohol/substances, and for benzodiazepines, which were more commonly used by patients who tested positive for alcohol/substances (Table 2). LAI antipsychotics are widely used in the treatment of schizophrenia [38]; the literature highlights that substance abuse is very common among schizophrenic patients who present to the ED [33,39]. This observation seems to contradict data from other studies. However, a more specific characterisation of these patients,

Their diagnoses and their symptoms would be necessary to fully understand this. Our study found no statistically significant correlations when considering these variables mentioned (Table 2).

### 4.3. Symptomatology Observed During the Psychiatric Evaluation in the ED

In both groups, the predominant symptomatology consisted of anxiety or mood disturbances (Table 3). This finding is not surprising when compared with the existing literature, as these are non-specific symptoms commonly observed in patients with a wide range of psychiatric diagnoses, including those more frequently associated with substance use behaviours, such as personality disorders. These symptoms are also among the most frequent in situations of personal crisis, which may not necessarily be linked to psychiatric disorders but may nonetheless be one of the main reasons why patients presenting to the ED receive psychiatric [40].

It should be noted that these symptoms were observed at a higher rate in consultations for patients who tested negative for alcohol/substances (Table 3). It is possible that within this group, there were patients experiencing alcohol withdrawal, which may be characterised by a negative toxicological screen but accompanied by anxiety symptoms [41].

Psychotic symptoms and cognitive impairments were also found to be more frequently observed in consultations for patients who tested negative for alcohol/substances (Table 3). These findings appear to partially contradict the existing literature, which indicates that psychotic symptoms, along with mania and delirium, are commonly associated with intoxicated patients. This discrepancy may suggest that other underlying factors could be contributing to the presentation of these symptoms in patients who test negative, warranting further investigation [2,42]. To gain a clearer understanding, it would be desirable to collect a larger dataset and provide a more detailed characterisation of these symptoms. For the same reasons, it is unsurprising that a negative psychiatric examination was more frequently observed in consultations for patients who tested negative for alcohol and substances. Psychomotor agitation showed a consistent distribution across both groups (Table 3), in line with existing literature.

### 4.4. Suicidality and Self-Harm

Additionally, both suicidal ideation and evidence of self-harming behaviours were more frequently reported in consultations for patients who tested positive for alcohol/substances (Table 4), consistent with findings in the literature [2]. On this topic, it is essential to study and specifically differentiate the behaviours exhibited, as the methods and motivations behind self-harming actions tend to vary depending on several factors, including diagnoses, present symptoms, and alcohol or substance intoxication (which can serve as a means of enacting self-harm). A deeper understanding of these distinctions is crucial, as the interplay between these factors can significantly influence both the presentation and underlying drivers of self-injurious behaviours, ultimately guiding more effective interventions and tailored treatment strategies [34]. In our study, no statistically significant correlations were found between the modes of self-harm behaviours and consultations for patients who tested positive or negative for alcohol/substances (Table 4).

### 4.5. Therapy Choice in ED

The administration of acute therapy was primarily necessary for consultations for patients who tested positive for alcohol/substances. In more than half of these cases, the treatment of choice involved intravenous or intramuscular administration of neuroleptics and benzodiazepines (Table 5). These medications were selected for their anxiolytic and sedative effects, combined with the rapid onset and efficacy of the injectable route, making them particularly effective in managing anxiety and psychomotor agitation [32]. Other treatments used included oral neuroleptics and benzodiazepines, with this route of administration being more commonly employed in consultations for patients who tested negative for alcohol/substances (Table 5). These findings are consistent with existing literature, suggesting that oral administration is typically preferred in less acute cases where rapid sedation is not as critical [28]. 

Lastly, the high prevalence of benzodiazepine therapy in consultations for patients who tested positive can be explained by the frequent use and abuse of this class of drugs among individuals with substance dependencies [43,44].

### 4.6. Outcomes of Psychiatric Consultation

In both samples of consultations, the intervention resulted, in most cases, in voluntary hospitalisation in Psychiatric Diagnosis and Treatment Service (PDTS) in the General Hospital or brief intensive observation, followed by discharge and referral to the Mental Health Center (Table 6). These data do not agree with what is reported in the 2022 Mental Health Report by the Italian Ministry of Health, which states that only 13.8% of total emergency room visits for psychiatric issues result in hospitalisation, with more than half being admitted to PDTS, while 72.3% result in home care [45]. The greatest differences were observed in the context of referrals to Substance Addiction Services (SerD) or any other specialists and Hospitalisation in compulsory healthcare treatment, where higher percentages were recorded in consultations for patients who tested positive for alcohol/substance (Table 6): in fact, these are patients who more often have dependencies and symptoms of acute intoxication that may require compulsory health treatment, given their poor cooperation and awareness of the illness, with follow-up at the SerD [2].

### 4.7. Study Limitations

The variable of alcohol/substance intoxication refers to the detection of alcohol levels and/or substances in the urine, including the presence of amphetamines, cannabinoids, cocaine, opiates, as well as barbiturates and benzodiazepines. This latter may not be abused by patients but could have been administered by ED physicians to manage psychomotor agitation, either in the field or upon arrival in the ED, before the drug test was performed. This could have increased the total number of intoxicated patients.

However, it is important to emphasise that at our centre, there is a great collaboration with the ED staff. It is known that, at the national level, toxicological screening is often not performed before psychiatric evaluation, posing significant risks to patient safety. Another limitation is that a complete profile of intoxications and dependencies could not be established, as many substances of abuse, particularly new psychoactive substances, are not routinely tested for in the ED, leading to the potential for false negatives.

Finally, it must be considered that in the COVID-19 pandemic, ED visits dropped significantly (nearly 42% for non-COVID issues) due to fear of infection, lockdowns, and resource reallocation. Conversely, COVID-19-related visits increased. This disparity led to concerns about delayed care for critical conditions, potentially resulting in worse health outcomes and prompting public health campaigns encouraging timely medical attention.

## 5. Conclusions

This study was conducted over three years, allowing for longitudinal analysis and the collection of data from 1362 psychiatric consultations carried out at the ED of Maggiore della Carità University Hospital in Novara, Italy, involving a total of 922 patients.

Even if this study was conducted in a single centre, the “Maggiore della Carità” University Hospital is the main referral hospital for all of northeastern Piedmont and the second dimension in Piedmont; its catchment area can be considered representative of the whole region. While it provided comprehensive data regarding the phenomenon in the city of Novara and its province, it would be beneficial to extend this research to a regional or national level to increase the sample size and assess the different challenges associated with this phenomenon on a broader scale, over a longer period.

This study aims to identify the characteristics of patients in psychiatric consultations at the ED, focusing on those who tested positive or negative for alcohol/substances, and to explore potential management strategies for addressing alcohol and substance abuse.

As expected, differences were observed between the two groups regarding both socio-demographic and clinical variables. This proved particularly important in two areas: the first pertains to the psychiatric diagnosis and symptoms that patients exhibited at the time of the consultation; the second concerns suicidality, with a major focus on the reasons and methods of suicide attempts or suicidal ideations. This would provide more effectively analyzable data to be compared with others from studies with a similar focus.

The limitations identified in the study also underscore the need for improvements in the ED to ensure better diagnostic classification of patients, particularly by increasing the number of substances that can be detected through blood and urine tests.

## Figures and Tables

**Table 1 ijerph-22-00023-t001:** Socio-demographic and clinical characteristics.

		Consultations for Negative Patients (Alcohol/Substances)	Consultations for Positive Patients (Alcohol/Substances)	*p*-Value
Gender	Male	43.30% (197)	50.83% (215)	0.025 *
	Female	56.70% (258)	49.17% (208)	
Who patients live with	Alone	22.19% (87)	20.06% (66)	0.019 *
	Family	69.39% (272)	67.17% (221)	
	Therapeutic structure/Prison	7.65% (30)	8.51% (28)	
	Homeless	0.77% (3)	4.26% (14)	
In charge of SER.D.	Yes	3.07% (12)	13.97% (50)	<0.00001 *
	No	96.93% (379)	86.03% (308)	
In charge of psychiatric service	Yes	65.24% (289)	56.03% (223)	0.006 *
	No	34.76% (154)	43.97% (175)	

* If the *p*-value is less than 0.05, it is judged as “significant”.

**Table 2 ijerph-22-00023-t002:** Ongoing therapy before the consultation.

		Consultations for Negative Patients (Alcohol/Substances)	Consultations for Positivee Patients (Alcohol/Substances)	*p*-Value
Typical antipsychotics (oral)	No	87.12% (399)	86.71% (372)	0.858
	Yes	12.88% (59)	13.29% (57)	
Tiypical antipsychotisc (LAI)	No	91.70% (420)	97.67% (419)	<0.0001 *
	Yes	8.30% (38)	2.33% (10)	
Typical antipsychotics (oral)	No	72.49% (332)	74.13% (318)	0.582
	Yes	27.51% (126)	25.87% (111)	
Atiypical antipsychotisc (LAI)	No	94.54% (433)	96.04% (412)	0.295
	Yes	5.46% (25)	3.96% (17)	
Antidepressants	No	69.00% (316)	67.37% (289)	0.602
	Yes	31.00% (142)	32.63% (140)	
Benzodiazepines (BDZ)	No	58.52% (268)	51.75% (222)	0.043
	Yes	41.48% (190)	48.25% (207)	
Mood stabilisers	No	91.48% (419)	91.61% (393)	0.947
	Yes	8.52% (39)	8.39% (36)	
Anticonvulsivants	No	95.41% (437)	96.27% (413)	0.524
	Yes	4.59% (21)	3.73% (16)	
AnticholinergicsAntihistamines	No	98.03% (449)	98.60% (423)	0.513
	Yes	1.97% (9)	1.40% (6)	
Other	No	96.94% (444)	97.20% (417)	0.819
	Yes	3.06% (14)	2.80% (12)	

* If the *p*-value is less than 0.05, it is judged as “significant”.

**Table 3 ijerph-22-00023-t003:** Symptomatology observed during the psychiatric evaluation in the ED.

Symptoms	Consultations for Negative Patients (Alcohol/Substances)	Consultations for Positive Patients (Alcohol/Substances)	*p*-Value
Psychomotor agitation	19.38% (88)	19.34% (82)	<0.0001 *
Mood disorders	17.40% (79)	11.79% (50)	
Intoxication	2.42% (11)	22.41% (95)	
Psychotic symptoms	18.50% (84)	14.5% (60)	
Anxious state	42.29% (192)	32.31% (137)	

* If the *p*-value is less than 0.05, it is judged as “significant”.

**Table 4 ijerph-22-00023-t004:** Suicidality and self-harm.

		Consultations for Negative Patients (Alcohol/Substances)	Consultations for Positive Patients (Alcohol/Substances)	*p*-Value
Suicidal ideation	No	83.85% (379)	77.25% (326)	0.014 *
	Yes	16.15% (73)	22.75% (96)	
Self-harming behaviours	No	83.44% (378)	74.53% (316)	0.001 *
	Yes	16.56% (75)	25.47% (108)	
Suicidal intention	No	70.83% (51)	62.39% (68)	0.241
	Yes	29.17% (21)	37.61% (41)	
Short-circuit behaviour	No	33.80% (24)	34.86% (38)	0.884
	Yes	66.20% (47)	65.14% (71)	
Self-harming behaviours	Defenestration/hanging	5.56% (4)	3.77% (4)	0.97
	Cut wounds	25% (18)	23.58% (25)	
	Ingestion	52.78% (38)	53.77% (57)	
	Other	16.67% (12)	18.87% (20)	

* If the *p*-value is less than 0.05, it is judged as “significant”.

**Table 5 ijerph-22-00023-t005:** Therapy choice in ED.

		Consultations for Negative Patients (Alcohol/Substances)	Consultations for Positive Patients (Alcohol/Substances)	*p*-Value
Therapy	Necessary	49.01% (222)	60% (255)	0.001 *
	Not necessary	50.99% (216)	40% (170)	
Kind of administered therapy	Other	5.43% (12)	11.07% (28)	0.001 *
	Antipsychotics/BDZ (intravenous/intramuscular)	48.87% (108)	53.75% (136)	
	Antipsychotics/BDZ (oral)	28.05% (62)	13.83% (35)	
	Polytherapy	17.65% (39)	21.34% (54)	

*** If the *p*-value is less than 0.05, it is judged as “significant.

**Table 6 ijerph-22-00023-t006:** Outcomes of psychiatric consultation.

		Consultations for Negative Patients (Alcohol/Substances)	Consultations for Positive Patients (Alcohol/Substances)	*p*-Value
Outcome of psychiatric consultation	Hospitalisation in compulsory healthcare treatment	9.21% (42)	12.80% (54)	0.0244 *
	Voluntary hospitalisation in Psychiatric Diagnosis and Treatment Service (PDTS)/Brief intensive observation	41% (187)	36.30% (153)	
	Reference to mental health centre	17.10% (78)	14.70% (62)	
	Reference to Substance addiction service (SerD)/another specialist	9.20% (42)	15% (63)	
	Hospital discharge	23.50% (107)	21.10% (89)	

*** If the *p*-value is less than 0.05, it is judged as “significant”.

## Data Availability

The data presented in this study are available on request from the corresponding author due to privacy rules.

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
