# Peer review of "Socio-Demographic and Clinical Characteristics of Patients with Substance Intoxication Receiving a Psychiatric Assessment in the Emergency Department of the Maggiore Della Carita Hospital, Novara, Italy"

_ijerph, 2024, doi:10.3390/ijerph22010023_

Round 1

Reviewer 1 Report

Comments and Suggestions for Authors

Check the attached document

Author Response

THE TITLE

Comments 1: Kindly rephrase the topic to be short and clear

Response 1: We adjusted the title to better align with the research topic. The new title is “Socio-Demographic and Clinical Characteristics Patients with substance intoxication received a psychiatric assessment in the Emergency Department of the Maggiore della Carita Hospital, Novara, Italy”.

Comments 2: I am suggesting this because alcohol is one of the substances, and the focus aspects could be included in the

Response 2: Agree. We have, accordingly, modified title to emphasize this point.

ABSTRACT

Comments 1: Kindly align the aim with the topic.

Comments 2: Keywords should be logical.

Responses 1 & 2: Thank you for pointing this out. We agree with this comment. Therefore, we have aligned the aim with the topic and choose more logical keywords.

INTRODUCTION AND BACKGROUND

Comments 1: Authors should add some more literature for a clear understanding of the seriousness of the problem

Comments 2: Kindly indicate the problem globally, Sub-Saharan, Nationally, and internationally could be incorporated.

Responses 1&2: Thank you for your suggestions. It's insightful to delve into these aspects. We have included additional literature to better grasp the significance of the issue and have integrated perspectives from international, Sub-Saharan, and Italian contexts.

MATERIALS AND METHODS

Comments 1: Kindly indicate the approach used and the designs

Response 1: We appreciate the feedback and valuable suggestions. We added the approach used and the designs.

Comments 2: Indicate if the pre-test was done. How many participants were involved in a pre-test? Were the findings of the pre-test included in the main study or not?

Response 2: A pre-test wasn’t done, even though pre-testing the data collection instruments, such as questionnaires, is essential to minimize errors and ensure reliable and valid data collection. Moreover, pre-tests may decrease the experiments' internal validity and internal validity by introducing a confound – the interaction between the pretest and the experimental manipulation. The pretest can interact with the experimental treatment, leading to effects that wouldn't have occurred without the pretest. This interaction makes it difficult to isolate the true impact of the independent variable. Pretesting has been avoided because of the time-consuming and resource-intensive. The pretest, in our case, is not specific to the research design. 

RESULTS

Comments 1: Indicate the reasons for choosing Chi-square/Fisher tests and t-tests or nonparametric alternatives instead of linear or multiple regression.

Response 1: Also, univariable and multivariable logistic models were performed to evaluate the association between the characteristics of patients with alcohol or substance intoxication and those without, considering socio-demographic variables and aspects of recent and past psychiatric history with a confidence interval set at 95%.

REFERENCES

Comments 1: Some references are older than ten years. Kindly use the recent references.

Response 1: Recent references have been added. Thanks for the suggestion.

Reviewer 2 Report

Comments and Suggestions for Authors

Research into substance abuse problems is of great importance. 

It would be advisable to include (briefly) the prevalence of substance abuse in Italy (as a context to the study). There few grammatical errors (e.g. line 44 and line 66) that may require editorial attention. 

It is advisable to have a section on ethical considerations (preferably under the 'Materials and methods' section) to demonstrate ethical integrity and rigor. Part of the information in line 308 could be incorporated. 

Author Response

Research into substance abuse problems is of great importance. 

Comments 1: It would be advisable to include (briefly) the prevalence of substance abuse in Italy (as a context to the study).

Response1: We agree with it. Consequently, we have added the prevalence of substance abuse in Italy. We believe putting this in the Introduction provides the additional context the reader requires.

Comments 2: There few grammatical errors (e.g. line 44 and line 66) that may require editorial attention. 

Response2: We agree that this needs to be carefully worded.

Comments 3: It is advisable to have a section on ethical considerations (preferably under the 'Materials and methods' section) to demonstrate ethical integrity and rigor. Part of the information in line 308 could be incorporated. 

Response3: Agree. We have, accordingly, modified 'Materials and methods' section to emphasize this point.

Reviewer 3 Report

Comments and Suggestions for Authors

The manuscript entitled "Patients with Alcohol or Substance Intoxication at the Emergency Department of the Maggiore della Carità University Hospital in Novara, Italy. Focus on Symptoms, Treatments, and Outcomes" aimed to portray the socio-demographic as well as the clinical profile of patients being treated in the Emergency Department and how their history of alcohol and/or substance abuse differentiates them from patients with no history of addiction. 

The manuscript is well-structured and well-written with certain results being of major clinical importance, such as the majority of female patients testing negative for alcohol/substance abuse compared to males, as well as the majority of homeless struggling with alcohol/substance addiction.

I recommend certain minor adjustments which should be made in order to additionally ameliorate the quality of the manuscript:

1) In the abstract it would be beneficial to state both the number of patients and psychiatric evaluations conducted in the ED.

2) In the study's limitations it should be included the phenomenon of the covid-19 pandemic and its potential impact on the attendance of patients to the ED.

3) I would like to propose a modification of the manuscript's title in order to be more concrete and attracting to the readers, such as: "Socio-demographic and Clinical characteristics of patients with Alcohol and/or substance dependance, attending the Emergency Department of the Maggiore della Carità University Hospital in Italy".

Author Response

The manuscript entitled "Patients with Alcohol or Substance Intoxication at the Emergency Department of the Maggiore della Carità University Hospital in Novara, Italy. Focus on Symptoms, Treatments, and Outcomes" aimed to portray the socio-demographic as well as the clinical profile of patients being treated in the Emergency Department and how their history of alcohol and/or substance abuse differentiates them from patients with no history of addiction. 

The manuscript is well-structured and well-written with certain results being of major clinical importance, such as the majority of female patients testing negative for alcohol/substance abuse compared to males, as well as the majority of homeless struggling with alcohol/substance addiction.

I recommend certain minor adjustments which should be made in order to additionally ameliorate the quality of the manuscript:

Comments 1: In the abstract it would be beneficial to state both the number of patients and psychiatric evaluations conducted in the ED.

Response1: On the reviewer’s recommendation, we have stated the number of patients and psychiatric evaluations conducted in the ED in the abstract section.

Comments 2: In the study's limitations it should be included the phenomenon of the covid-19 pandemic and its potential impact on the attendance of patients to the ED.

Response2: We agree with this comment. In the limits section, we included the phenomenon of the covid-19 pandemic and its potential impact on the attendance of patients to the ED.

Comments 3: I would like to propose a modification of the manuscript's title in order to be more concrete and attracting to the readers, such as: "Socio-demographic and Clinical characteristics of patients with Alcohol and/or substance dependance, attending the Emergency Department of the Maggiore della Carità University Hospital in Italy".

Response3: We agree with this and have incorporated your suggestion throughout the title of the manuscript.